# Reassessing the Inculcation of an Anti-Racist Ethic for Christian Ministry: From Racism Awareness to Deconstructing Whiteness

**Anthony Reddie** [1,2] 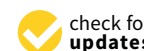

1  Regent's Park College, The University of Oxford, Oxford OX1 2LB, UK; anthony.reddie@regents.ox.ac.uk
2  The Department of Philosophy, Practical and Systematic Theology, The University of South Africa, Pretoria 0002, South Africa

**Abstract:** This paper outlines the means by which candidates training for Christian ministry are encouraged to engage with the deontological positionality of anti-racism as a substantive element of Christian praxis. The first part of the paper provides some brief historical reflections on what was then the conventional approach to teaching an anti-racist ethic for Christian ministry, namely, the practice of "racism awareness". Following these reflections, the author proceeds to outline the epistemological change that has occurred in his own ethical teaching, moving from the focus on racism awareness to a more critical, postcolonial deconstruction of Whiteness and its concomitant links to Mission Christianity. Mission Christianity, the religion that underpinned the British Empire, is identified as the repository that helped to institutionalise the existence of "white supremacy" and racism within the body politic of colonialism and the rise of notions of "manifest destiny". In switching the modus operandi for an anti-racist ethic within Christian ministry, this paper seeks to reframe the ways in which the ethical basis for opposing and resisting racism is effected within Christian theology

**Keywords:** anti-racist ethic; Mission Christianity; racism awareness; Whiteness; Christian ministry

## 1. Historical Background

My scholarly development has always existed in a series of dialectics. The primary ones of relevance in this paper are the constructive tensions between scholarship and ministry and Practical theology and Black theology. In terms of the latter, my scholarship has sought to utilise radical, liberative models of transformative education[1] as a conduit for undertaking Black theology.[2] This thematic and methodological form of scholarly engagement is for the ultimate purpose of the conscientisation and Christian formation of predominantly laypeople for the purposes of radical, anti-racist forms of Christian discipleship. My participative approach to undertaking Black theology is one that seeks to use models of experiential learning, such as exercises and games, role-play and drama, as an interactive means of engaging with adult learners in order that they can be impacted by, learn from, and contribute to the development of new knowledge concerning the theory and practice of Black theology.[3] Whilst

---

1  Transformative Education is a form of knowledge construction that challenges the dominant theories and paradigms that constitute normative frames of epistemology. It proceeds from a critical, dialectical inquiry into the very basis of what constitutes knowledge and truth. One of the most instrumental texts in my own intellectual development is (Banks 1996).
2  Black theology is a theology of Liberation whose point of departure is the existential struggles of Black people of African descent seeking to interpret their lives' experiences in dialogue with the God, revealed in Jesus Christ, whom they believe represents the source and framework for their attempts to resist oppression and marginalization. The first Black theology text was (Cone [1969] 1986). See also (Reddie 2012).
3  Extensive examples of this work can be found in (Reddie 2005, 2006a, 2008).



the bulk of this participative research has been focused on engagement with predominantly Black laypeople, an alternative strand has focused on working with predominantly White people training for ordained ministry.

This form of pedagogical and research engagement once took the form of "racism awareness raising workshops"[4] for predominantly White theological education and ministerial training students in various institutions across the country. This work took the form of being a consultant in Black theological studies for the Methodist church[5]. The term "race" is in inverted commas as in strict terms, it does not exist. The notion of "race" as a set of unproven frameworks for indicating the notion of fixed categories of biological (and hierarchical) differences between differing groups of people is an invention or fiction of the era of modernism[6]. Racism is the outworking of the concept of "race", which results in behavioural, procedural, policy, systemic and practical forms of discrimination based on the prejudices and stereotypes that arise from this construct.[7]

This essay is not seeking to offer a substantive genealogy on the ongoing constructive discourse surrounding the phenomenology of "race". Rather, it seeks to synthesise aspects of this continuous discourse for the specific purpose of rethinking the current discourse and praxis of ministerial training as pertains to inculcating an anti-racist ethic. In effect, this work is less about the specifics of "race" as a conceptual category and is more concerned with the application of the latest thinking on this phenomena as a theological problem, particularly as it pertains to theological education and ministerial training.

## 2. Racism Awareness

"Racism awareness training" has become an ingredient within British theological education over the last forty or so years. This form of training in theological education was pioneered by the British Methodist church, arising out of the landmark report *A Tree God Planted: Black People in British Methodism*,[8] in which the term "institutional racism" was first coined within the context of Church life, some fourteen years before the government instituted the Macpherson report that looked into the death of Stephen Lawrence, which gave legitimacy to the concept and the term.[9]

Racism awareness training was for many years compulsory for all persons training for public authorised ministry within the Methodist church and in many of her ecumenical partner churches.

---

4    "Racism awareness raising" approaches to ministry arose out of the wider context of anti-racism approaches to experiential models of informal education and learning that seek to inculcate within participants an increased understanding and a heightened awareness of the existence and ramifications of oppressive forms of action based on the construct of "race". Racism awareness training first emerged in the UK as a form of anti-oppressive training in professional development in occupations such as housing, social work, youth and community work, and probation work. For more details on this, please see (Tamkin et al. 2003).

5    Between 1 September 1999, and 31 August 2010, I worked as a consultant in Black theological studies, based at the Queen's Foundation for Ecumenical Theological Education in Birmingham. This role involved working part time while undertaking postdoctoral research in Black theology and Christian education based at Queen's. The other "half" of the role saw me travelling across Methodist theological institutions and ecumenical ones in which Methodist students' training for authorised, public ministry is located, working as a theological consultant. In working as a consultant, my role was to work as a guest lecturer, teaching Black theology, postcolonial theologies, Liberation hermeneutics and, most typically, leading "Racism Awareness Days". The latter were for the purpose of conscientising ministerial students, equipping them to be anti-racist practitioners in the respective ministries.

6    A number of scholars have demonstrated the specious nature of such discourse and the ways in which it seeks to create untenable and unstable boundaries between groups of humanity. The concept of "race" is a misnomer as it strictly does not exist, as there is only one "race", the human race. Racism, however, does exist, as forms of discriminatory action based on the grounds of "race" are very real. See (Barndt 2007) and (West 2002). See also (Hopkins 2005). See also (Carter 2008; Jennings 2011; Douglas 2015; Eze 1997).

7    For an excellent distillation of the construct of "race" and racism in the context of Christian discipleship and ministry, see (Ackroyd et al. 2001).

8    See (Walton et al. 1985).

9    Stephen Lawrence was a seventeen-year-old Black African Caribbean young man who was brutally murdered on the streets of South London in a frenzied and unprovoked racially motivated attack. The Metropolitan police in London (the overarching authority for investigating crime and policing the streets of London) was roundly attacked in the media and within Black communities in Britain for their racist-inspired neglect and inefficiency in investigating the case. No one was initially convicted for Stephen Lawrence's murder. The public outcry in response to the negligence of the Metropolitan police service gave rise to the MacPherson report. For further details on the Macpherson report, see (MacPherson 1999).

This training sought to conscientise ministerial students to become knowledgeable on the historical manifestation of the conceptual idea of "race" and its dangerous offspring, namely, "racism". This form of enacted pedagogical practice was undertaken by means of informal workshops over the course of one- or two-day teaching and learning events. These were described as "Racism Awareness Days", although they were also termed "Racial Justice Training Days" or "Exploring Anti-Oppressive Practice". The emphasis of this was not simply to assist predominantly White students to become "more aware" of the conceptual frameworks of "race" and racism. Rather, the learning outcomes of these days focused on developing anti-racist models of ministry in order to assist students in combatting and resisting racism in all its forms. The underlying ethical basis of these training days was to challenge White ministerial students to develop action plans for the enactment of praxiological, anti-racist models of Christian ministry that would see such individuals acting as "White allies" in supporting the work of anti-racism in the church and beyond.

My involvement was often that of a guest lecturer, charged with engaging with a group of predominantly White students training for ordained ministry in a variety of theological institutions, from mainly the Methodist, Anglican, Baptist, and URC denominations. These sessions were focused on "theological anthropology", inviting participants to reflect on the nature of their identity; namely, what are the constituent parts of their human constructions?

In this pedagogical approach to "Black participative theology", I used a variety of exercises and activities for enabling participants to explore their feelings and emotions in a safe space. The exercises allowed them to adopt imaginary roles and to "park" their sometimes "extreme" feelings within a comparatively safe "rest area" where they could notionally ascribe responsibility for their anger, frustration, or sense of tension to the fictional persona of the character they had adopted in the exercise.

A participative approach to Black theology, linked with transformative Christian education, for the purposes of encouraging adult learners to engage in anti-racial models of Christian discipleship and ministry, is one that uses Martin Luther King Jnrs notion of the "Beloved Community"[10] as its central heuristic. The use of exercises and drama represents an invitation for adult learners to reflect within the hospitable and safe space of the workshop. In this context, they can explore and commit themselves to working for and becoming a part of the collective spiritual and psychological journey of the Christian church, towards the "promised land" of racial justice, what, in effect, I would describe as the "reign of God" or "Gods gracious economy".[11]

In the various exercises, participants, by means of conversation and interaction, have the opportunity to reflect on their actions within the context of a central activity and to assess their agency and responses to it for their truthfulness to Gods gracious activity in Jesus Christ when juxtaposed with the historical and contemporary experience of racism and oppression.[12]

The participative element of the work challenges learners to decide how they will inhabit particular spaces and places in order to assess the ways in which they are playing out learnt pathologies that are often informed by the specious binaries of "them" and "us".[13] What would happen if participants were enabled to take on the persona of the "Other" in order to live out their realities and experiences within a participative exercise? To what extent would these experiences change their subjective self and their concomitant consciousness? Ultimately, in what ways would the resultant change in consciousness inform their future praxis as ministers in public, authorised ministry?

---

[10]　See (Baldwin 1995).

[11]　I am very much indebted to my friend and colleague Michael N. Jagessar for the latter term or phrase for naming the eschatological hope of God's justice and equity for all persons that constitutes our futuristic hope within the Christian faith. See (Jagessar 1997).

[12]　See (Andrews 2014, pp. 11–29).

[13]　African American Womanist theologian Kelly Brown Douglas demonstrates the extent to which binary notions of "in groups" and "out groups" within Christian communities can be traced back to the notion of a "closed monotheism" within Judaeo-Christian theologies of the Hebrew Bible and the New Testament. See (Douglas 2005).

The journey towards the beloved community is one in which the process is as important as the destination that is reached. In the context of performative action, one is constantly challenging participants to question their assumptions about what is deemed to be "normative" and that which is termed as "aberrant" or "transgressive". The modus operandi of this approach to undertaking participative Black theology is for the purpose of offering participants new models of being Christian in a context where White nationalism and racism are on the rise.[14]

The purpose of this approach to undertaking Black theology lies in the belief that internalised change (spiritual and psychological) can be a conduit for externally verified changes in behaviour and practice. Both of these modes provide the subjective, experiential basis for liberation at an individual, interpersonal, communal, and ultimately systemic level.

This model of liberative, pedagogical work is predicated on a participative teaching and learning process. The natural corollary of this pedagogical approach is a model of liberative theological reflection that is undertaken by means of participative exercises through which new theories and concepts for Christian praxis are enacted.

It is my belief that Christian ministry remains the central context in which ordinary Christians and ministers can seek to use the insights of Black theology in order to become signs of hope and models of change for the liberation of all people who are presently marginalised and oppressed. Whilst building on my past work, these sessions, nevertheless, represent something of a slight departure from my previous scholarship.

The problems with racism awareness training were many, but in brief, I will summarise these ways. First, in its operation as a stand-alone, non-assessed initiative, the work was always envisioned as an atypical form of educational provision separate from the substantive curriculum work of ministerial training and theological education. In the failure to integrate racism awareness into the substantive heart of ministerial training and theological education, the work of anti-racist educators, such as myself, was fatally undermined, as these initiatives were being perceived as barely tolerable encumbrances to the "serious" and "proper" work of ministerial formation.

I remember one memorable occasion when leading a racism awareness training day in a particular Methodist institution during which a ministerial student verbally abused me and other members of the class, kicked over tables, and stormed out of the room in anger and disgust at my teaching. When the student returned, unrepentant and unreflective on their antisocial and wholly inappropriate behaviour, I challenged them by saying, "if you can tell me at what point in your future pastoral ministry your behaviour would ever be acceptable, then I will shut up and never teach these days again." This ministerial student walked out again and did not return.

I mention this incident because my assumption was that this extreme, inappropriate behaviour might be an impediment to the individual being accepted into the Methodist ministry, but it was not. In fact, my opinion was not sought in terms of the fitness of the person for ministry, because by inference, my running of this atypical, non-assessed, one-off adjunct form of enterprise was simply a tick-box exercise that had no major bearing on the formation of candidates for ministry.[15] Showing distain for and behaving badly in the one educational input focused primarily on racism and White supremacy, was not an impediment to their being engaged in authorised, public Methodist ministry. This epistemological lacuna reminds me of the great charge James Cone has levelled against White, Euro-American theological ethics.[16] Namely, that Christian theology has developed a penchant for observing in minute details many forms of theological abstraction, but has largely refused to observe

---

[14] See the following website where the then leader of UKIP, Nigel Farage, clearly invokes a cultural interpretation of Christianity as a means of promoting a reactionary, homogeneous construct of Britain. http://www.secularism.org.uk/news/2015/04/nigel-farage-calls-for-muscular-defence-of-christianity-in-the-uk (National Secular Society 2015). Journalists have noted the rise in racist attacks and right wing nationalism. See (Booth 2019) for further details.

[15] At the time of writing, these "one-off days" were non assessed and were not integrated into the main curriculum of theological training and formation. This situation may no longer be the case.

[16] See (Cone 2004).

the visceral and palpable nature of racism within the body politic of White majority societies and the churches located within them.

The second problem with these days was the failure to ground them in the epistemological heartland of the theological curricula that underpinned the ministerial training and formation of Methodist ministers. By focusing on the concept of "race" and concomitant manifestations of racism, as opposed to the underlying frameworks of the theological construction of White Western Christianity, this approach enabled the reification of the opacity around the continued flourishing of White supremacy. In focusing on abstractions of "race" separated from the centrality of Christian theology as expressed within the curriculum of theological training, this led to the continued diminution in the veracity of racism awareness training and the significance with which it was perceived as an essential component in the ontological development of ministerial students training for ordained ministry.

In critiquing Racism Awareness, I need to acknowledge the limitations of my own practice. The institutional and epistemological frameworks I have described were ones I readily agreed to work within, in that I agreed with the belief such forms of operative pedagogy were entirely reasonable. Working as a Practical Black theologian, I saw this work as an adjunct to my more normative work as a theological educator and not a trainer and facilitator for racism awareness in stand-alone, one-off, day-long activities.

## 3. Unconscious Bias and Equalities, Diversity, and Inclusion

In the years since this work came to an end, the emphasis has moved on from an analysis of racism to one of "unconscious bias". Unconscious bias is a social identity theory that seeks to enable individuals to deconstruct their embedded world views and the ways in which these impact on their perceptions and outlooks as they engage with others in the world as historical subjects.[17] The shift in culture in how an anti-racist ethic is inculcated within the wider formation for ministry within the Methodist church has coincided with the shift in the corporate policy, moving from anti-racism or "racial justice" to one of equalities, diversity, and inclusion (EDI) and moving from the Queen's Foundation in Birmingham to the Susanna Wesley Foundation, based within Roehampton University[18] in South London. The national policy of unconscious bias has been ensconced within a research project that seeks to rethink models of diversity and inclusion within the Methodist church and how notions of difference are handled within the context of Wesleyan ecclesiology.[19]

This shift from a Black-theology-inspired ethic for anti-racism to one of unconscious bias has, as I will demonstrate in the final section of this article, serious implications for how we understand the challenges posed by systemic racism that have been revealed by the Coronavirus pandemic and the death of George Floyd, leading to the resurgence of the Black Lives Matter movement. Does the individualisation and ahistorical, epistemological framing of unconscious bias enable participants to come to terms with and deconstruct the worst excesses of White supremacy and the privileging of Whiteness that has underpinned the concept of "race" and the realities of racism over the past 500 years?

It is important to assert the significant impact of unconscious bias as a form of transformative learning pedagogy that seeks to conscientise participants in a variety of social and institutional settings.

---

17   For the basics of "unconscious bias" and its related training, see (Wikipedia 2018). Unconscious bias training continues to use experiential and progressive modes of pedagogy that seek to enable individuals to access their affective domain as a means of instituting behaviour change. In supporting individuals in responding to their emotional states, as the means by which substantive consciousness-raising modes of change can be enacted, this form of pedagogy relates to the broader developments in transformative education and learning in which my work has been located. In short, I am not suggesting any substantive divergences in the pedagogical approaches of my previous work in racism awareness and the contemporary use of unconscious bias forms of pedagogy within the Methodist church. Rather, my critique lies in the epistemological underpinnings of the latter and the developing approach to anti-racism that I am detailing in the second half of this paper.
18   For details on the Susanna Wesley Foundation, see (The Susanna Wesley Foundation 2020).
19   One can see examples of this in the research project that has given rise to the main outputs detailed in (The Susanna Wesley Foundation 2019).

My critique of unconscious bias does not extend to the utility and efficacy of this philosophical and pedagogical approach in principle. Although there are queries directed at the effectiveness of "unconscious bias" training[20] in terms of diversity and equalities strategies as they pertain to corporate management, my concerns with its utility are located solely in terms of inculcating ethical forms of ministry as it relates to Christian ministry.

My problem with this new model is the lack of any serious analysis of the wider socio-cultural and political construction of Empire and the ways in which the embedded nature of Whiteness has formed a world in which notions of manifest destiny and White exceptionalism have given rise to a toxic reality built on White supremacy.[21] African American Black religious scholar Stephen Ray has demonstrated how the construct of Whiteness amongst the White settler colony of the US, building on their European roots, reifies the means by which White Christianity represents the sublimated superstructure that underpinned the ethical basis of modern America.[22] The conflation of the cross, White supremacy, and notions of White manifest destiny creates the theological frameworks that enshrine Whiteness as the regulatory norm for what constitutes righteousness and belonging.[23]

The individualising of the tenets of unconscious bias enables the wider systemic means by which Whiteness constructs a socio-cultural and political platform on which White supremacy is enacted to go unnoticed. African American Womanist ethicist Emilie Townes has written about the cultural construction of evil via the media depictions of Blackness and how the toxic ephemera of the media exacerbates this hegemonic dynamic of White supremacy.[24] Townes' penetrating analysis of the cultural production of Western life reminds us of the embedded ways in which Blackness is fixed in the popular imagination.[25] The wider constructs by which Whiteness has enveloped the Christian project are summarily ignored as the church focuses on individual oversights and wrestles with ways in which visible minorities might be integrated into the White socio-cultural framing of normality. In 2020, can it really be the case that a religio-cultural framing for an ethical approach to justice-making should adopt as its modus operandi a focus on integrating minorities as opposed to deconstructing inherited, systemic, White epistemological norms? Conversely, my own developing work draws on the brilliant insights of Willie James Jennings. Jennings explores the construct of "race" within the body politic of Christianity in exemplary fashion using several generative stories of how the world of Europeans collided with that of Africans, and it is in this combustible nexus that the new, toxic order of Christian thinking emerges.[26] The creation of this alternative approach to creating an anti-racist ethic for those training for Christian ministry is predicated on a critical rereading of Christian tradition and the concomitant development of White Eurocentric theology.

This developing work I am describing is an acute critique of the racism awareness work I used to undertake and its successor that is framed within the intellectual frameworks adopted by unconscious bias training.

## 4. Telling an Under-Told Story: The Role of Christianity in Creating Anti-Black Racism

The development of an alternative pedagogy for effecting an anti-racist ethic for Christian ministry commences with a historical deconstruction of the role of White Christianity in the Transatlantic slave trade. I am arguing that there existed (and continues to this day) an underlying framework that

---

20    For details on some of the criticisms directed at unconscious bias training, see (Noon 2017, pp. 98–209).
21    See (Reddie 2019).
22    See (Ray 2010).
23    (Ray 2010).
24    See (Townes 2006).
25    (Townes 2006, pp. 11–55).
26    See (Jennings 2011).

enabled many Christian churches to construct an ideology, based upon an incipient, racist theology, that assisted them in supporting Black chattel slavery, which was unhindered by any faith in God.[27]

To understand the churches' role in slavery, we need to look back to the early Church Fathers and ideas derived from Greek Antiquity. It is in this much earlier period in the first four centuries of the "Common Era" (CE) that ideas of Black people as the negative "Other" first begin to surface in Christian thinking. The later period of European Expansion around the time of the Crusades and the violent conflict with African (Black) Moors (Muslims) lead to the intensifying of ideas around Christianity=Europe(Christendom)=White versus Non-Christians=Africa(Barbarians)=Black. Black people became the Other.[28]

The aforementioned is exacerbated by the fact that White Christianity *is a violent religion*. It is based upon a form of "closed monotheism"—i.e., the "Christian God" is a jealous and competitive God who will not tolerate rivals and the "Other" who worship such God(s),[29]—which in turn is conflated with White exceptionalism, privilege, and power. Therefore, the conflation of White Christianity and the hermeneutics of power leads to forms of aggressive social-political praxis that are often predicated on violence. This can be seen in a number of Hebrew Bible texts, in which a "competitive" God instructs the people of Israel to commit genocide on others who inhabit the "Promised Land"[30] (see the book of Exodus, Chapter 23, Verses 20–33 and the list of peoples overthrown in the book of Joshua, Chapter 12).[31]

In invoking the term "violent religion" in regard to White Christianity, I am speaking towards the wider Judeo-Christian tradition, which, when allied to notions of White supremacy, becomes the hermeneutical lens for rereading the aforementioned texts in Exodus and Joshua on which "Christian genocide" is enacted. This view is explicated in the work of Robert Warrior, who reflected on the manifest destiny of White settler communities in the US, whose use of the "closed monotheism" of Christianity enabled the justification for the usurping of Native American land.[32] Warrior's claim about the violent impulses of White Christianity that have helped to fuel White supremacy that underpinned European imperialism is amplified in the work of a number of international scholars and activists from the Global South. These scholars and activists have demonstrated the means by which White Christianity has been able to colonise the Judeo-Christian tradition in order to dominate and subjugate others, often people of darker skin, across the world.[33]

When you combine the questionable attitude to Blackness with the sense of competition with people who are not like you (i.e., Black), you have a potent cocktail for an underlying theology of "them" (Black people or the Other) and "Us". You know who the "them" are, because they do not look like you. They are not "of God" (not "His people") and therefore "all bets are off" in terms of how you treat them.

When European traders, particularly in the Elizabethan age, began to engage with Africans on a prolonged basis, mainly through trade, it did not take much imagination to see that the underlying notions of "Otherness" made Black Africans ripe for exploitation.[34] The tensions between religion, faith, ethnicity, and nationality are then exploited by means of "specious" Biblical interpretation—the main text that resolved the issue of justifying the enslavement of Africans within a Christian framework came from Genesis 9:18–25—The Curse of Ham. Noah punishes his son Ham by cursing his own

---

27　For an incisive and critical interrogation of the corruption of Christianity by notions of "race", which assisted in the theological construction of chattel slavery, see (Carter 2008).

28　See (Hood [1994] 2000).

29　(Douglas 2005).

30　For a critical rereading of the Exodus narrative, which offers an anti-imperialist, anti-hegemonic hermeneutic, see (Warrior 1997, pp. 277–85).

31　For a wider discussion on the destruction consequences of the book of Joshua, see (Bridgeman et al. 2010, pp. 180–88).

32　See (Warrior 1997, pp. 277–85).

33　See (Hopkins and Lewis 2014).

34　See (Gerzina 1995).

grandson Canaan (the son of Ham), condemning him and all his descendants to slavery.[35] Since there was a widely perpetuated belief that Africans/dark-skinned peoples were the descendants of Ham, this so-called "curse of Ham" was used as biblical evidence that the enslavement of African people was actually willed and sanctioned by God. There was also a similar but less well-known argument based on the biblical story of Cain and Abel (Gen 4: 8–16), where the "mark of Cain", punishment for the murder of his brother, is interpreted as representing Black skin. Again, people of African origin are somehow identified as cursed by God for some past wrong. Here, any notions of blame are removed from the slave owners, since it can be said that the condition in which the Africans find themselves as slaves is due to the sins their ancestors have committed in the past, for which God is punishing them. Their Black skin is seen as proof of their sinful condition.[36]

Proponents of the Atlantic slave trade constructed such wild and fantastical forms of interpretation of the Bible (in support of slavery) because of the presence of pre-existing views of Africans as "Other" and as being "cursed by God".[37]

The aforementioned was ameliorated after the Haitian revolution at the end of the 18th century. The charge to "Christianise" enslaved Africans was undertaken on a number of Biblical and theological terms. There was a dichotomy between the body and the soul. This dualism is a particular outworking of Pauline theology. Salvation is achieved solely by faith in Jesus Christ. In the theological construction of Pauline theology, salvation is not dependent on praxis, but on faith in the saving work of Jesus.[38]

This means that if you are a Christian slave owner, you can have faith in Christ and still own slaves, as God is only interested in your soul, which is preserved through faith in Jesus. Your actions on earth are another matter.[39] For the enslaved Africans, faith in this same Jesus guaranteed salvation in heaven but not material freedom here on earth for the same reason as that given for the justification of the actions of slave masters. In the theological construction of slave holding economies, Africans could be saved. Given that this underlying framework of European superiority still held sway, however, even when both Black and White were members of the same religious code (the Body of Christ), it is no surprise that after the abolition of the slave trade and later slavery itself, Europeans continued to oppress Africans. It is interesting to note that the "dash for Africa" in the mid 19th century came soon after slavery was finally abolished in the British Empire.

The existence of racism in Britain today, as we speak, is testament to the continuance of the underlying Eurocentric Judeo-Christian framework that has always caricatured Africans as "less than" and "the Other", i.e., not one of "us". So why am I still within the Christian church trying to effect an anti-racist ethic in Christian ministry? I remain a practising Christian because there is another story to be told. One that lies in the heart and mind of such luminaries as Sam Sharpe, a Baptist Deacon who initiated the largest rebellion in Jamaica against slavery, in the Christmas period of 1831. For Sharpe as well as other enslaved Africans, Jesus was the Liberator who came to bring freedom to the captives.[40] Texts like Luke 4: 16–19 or Matthew 25: 31–46 from the Gospels became "proof texts" that God as reflected in the life, death, and resurrection of Jesus was on the side of the oppressed and the suffering and against the perpetrators of the slave trade.

This Black radical tradition in Christianity continues in the present day. Black people have continued to re-interpret the meaning of Christian faith in order to challenge illegitimate White power (and Black power, also, when it should be called to account) and to proclaim freedom for all people, a freedom that speaks against the continued realities of racism and White supremacy.

---

35  See (Johnson 2004).
36  See (Hood [1994] 2000).
37  See (Douglas 2005).
38  See (Reddie 2007) for a more in depth analysis on this issue.
39  (Douglas 2005, pp. 150–98).
40  See (Reddie 2006b).

## 5. Deconstructing of Mission Christianity

Building on the critical rereading of Church history and Christian tradition, I want to locate the thrust for an anti-racist ethic as one that moves beyond the framing of my previous modalities of this work, as outlined in the first section of this article. This development has moved from a focus on the minutiae of delineating the formulations of "race" and the manifestations of racism in Britain and across the world to one that seeks to explore the internalisation of White supremacy within the phenomenological edifice that is Mission Christianity.[41] In using this term, I am speaking of a historical phenomenon in which there existed (and continues to this day) an interpenetrating relationship between European expansionism, notions of White superiority, and the material artefact of the apparatus of Empire. This form of Christianity became the conduit for the expansionist paradigms of Eurocentric models of Christianity in which ethnocentric conceptions of Whiteness gave rise to notions of superiority, manifest destiny, and entitlement.[42]

Central to the development of Mission Christianity has been the framing of Whiteness as the signifier for notions of righteousness and axiomatic tropes of regulatory patterns of Christian discipleship. James Perkinson demonstrates how Whiteness provides the superstructure on which Enlightenment rationality is predicated, with which Mission Christianity happily colluded as a means of constructing notions of normality versus deviance in the application of the faith across the "New World".[43] Perkinson argues that "Whiteness was born of the European encounter with people, places, and things that fit no clear category on the map of Christian cognition".[44]

The aforementioned work, in terms of both the racism awareness training I led or the later development of unconscious bias training I have named, for all its attempts at instituting a liberative pedagogical ethic, is nevertheless predicated on an assumption of the normativity of the Christian faith as an inviolate guarantee for non-racialised discourse and praxis. One cannot ignore the reality that the Bible has been used to justify slavery, colonisation, rape, and homophobic violence.[45] It has been used as a weapon against Black people. Ideological biblical scholars such Randall Bailey and Oral Thomas talk about the need to read against the text.[46] In fact, Bailey, who I would identify as the doyen of socio-political, ideological readings of the Bible, asserts the importance of an ideological mode of interpretation that is commensurate with one's own religio-cultural bias.[47]

Without the robust deconstructive work as I have described, there is a danger that the radical pedagogy underpinning racism awareness training, or the later iteration of an anti-racist ethic that is enshrined in unconscious bias training, effectively becomes a de facto White-controlled discourse much like the more centrist "classical theology" and Biblical studies that have shaped Western Christianity. One of the perennial problems with the racism awareness work I used to undertake lies in its reliance on the normative frameworks of Christian discourse and the failure to name the White, privileged, androcentric inherited norms that have underpinned Christianity since the epoch of colonialism. It has been this unexplored, embedded nature of Whiteness that has enabled Western Christianity to all too easily collude with racism and the subterranean tropes of White supremacy.[48]

The development of the Equalities framework within the national life of the Methodist church has been distilled within the Equalities, Diversity, and Inclusion (EDI) toolkit, in which the commitment to

---

[41] Mission Christianity is the form of Christian faith that went hand in hand with the British Empire. London Missionary Society evangelist and explorer David Livingstone is reputed to be the author of the infamous 3 Cs: Commerce, Christianity, and Civilisation. For further details, see (Nkomazana 1998).

[42] For a helpful dissection of this model of Christianity, particularly the British version of it, see (Gorringe 2004). See also (Hull 2014).

[43] See (Perkinson 2004, pp. 151–84).

[44] (Perkinson 2004, p. 156).

[45] The Bible was a key tool in justifying the enslavement of African people. Two amongst the many texts in this area are (Haynes 2002). See also (Johnson 2004).

[46] See (Bailey 1998, pp. 66–90; Bailey 2010, pp. 31–46). See also (Thomas 2010).

[47] (Bailey 1998, pp. 66–90).

[48] See (Cone 2004).

justice-orientated ministry is enshrined within the various units and workplans constructed by the connexional (national) committee.[49] Within the material developed in the EDI toolkit, one can see clear evidence of a commitment to wrestling with historic, theo-cultural frameworks of Whiteness and the embedded, historic constructs of White supremacy that have defined Christianity as the religion of Empire.[50] The toolkit has a section on "race" (module 6), which includes the generative work of Peggy McIntosh[51] that addresses White privilege and the normativity of Whiteness as an unmarked form of human subjectivity. Now, to be clear, McIntosh's pedagogical activism is hugely significant, and the inclusion of this work demonstrates a level of radicality that reveals the substantive ethical intent to address racism and White privilege within the body politic of the church and wider society.

The failure of the EDI strategy adopted by the Methodist church lies in its failure to engage with the radical "theological" deconstruction of Whiteness that is employed by James Perkinson. In his pioneering text, *White Theology*,[52] Perkinson, a White, Euro-American theologian, wrestles with the phenomenology of Whiteness in order to deconstruct the economic positionality of Whiteness and critically challenge the worst forms of theo-anthropological obfuscation that often underpin the existing modalities of anti-racism found in my earlier racial justice work or the more recent EDI framework adopted by the Methodist church. One cannot casually remove Christianity from the contaminating stain of Whiteness as if the problems of White supremacy exist solely beyond the parameters of the Christian faith itself and have not become embedded within the very epistemological framing of the phenomenon of Christianity across its history.

In wrestling with the colonial hinterland of Mission Christianity and the emblematic ways in which Whiteness is embedded within it, this developing approach towards an anti-racist ethic in terms of Christian ministry is one that seeks to help White people connect with their unnamed Whiteness. How does one enable White people to reflect critically on their developing Christian discipleship and ministry in a manner that is informed by their surreptitious and usually unstated Whiteness, which in Christian theology has been as much a hindrance as it has been a help?

In Christian theology, one often witnesses studious attempts to avoid engaging with embodied difference.[53] When I was an undergraduate student in Church History at the University of Birmingham many years ago, we spent a great deal of time looking at the writings of great luminaries such Martin Luther, John Calvin, et al. At no point were they ever racialised, i.e., ever described as "White authors" or "White thinkers". These individuals were simply "authors" or simply "thinkers". Their ideas were generic and most importantly, they had universal implications for all peoples.

Whiteness operates as an overarching construct, which assumes a central place in all epistemological and cultural forms of production, thereby relegating other positions or perspectives as "Other". I should, at this juncture, make the point that I am not seeking to traduce all White people, nor am I constructing this discourse on the pejorative understanding that Whiteness is aberrant or wholly without legitimacy. It is also important that I make the point that feminism and gender studies add a particular piquancy to this debate, because they contextualised and complicated the nature of this discourse, as Whiteness has to be contextualised in terms of other overarching vistas such as class, sexuality, patriarchy, and androcentrism.[54]

---

49　Details of the EDI Toolkit can be found in (The Methodist Church n.d.a.). The EDI Toolkit has become the new normative training resource for demonstrating the ethical approach to justice-making and equity within the Methodist church. The Toolkit is comprised of a number of modules that cover differing aspects of equalities, diversity, and inclusion from a legal and a faith-based perspective. Unconscious bias provides the underlying intellectual basis for the various exercises and units that comprise the Toolkit. In effect, there is a clear intellectual link between unconscious bias and the practical pedagogical resource that is the EDI Toolkit, which includes a plethora of fine and intelligent experiential learning resources that seek to conscientise participants in undertaking anti-oppressive forms of Christian ministry and activism.

50　See (Sugirtharajah 2003).

51　For details of this work, see (McIntosh 1990).

52　See (Perkinson 2004). See also (Perkinson 2005). See also (Cassidy and Mikulich 2007).

53　A classic example of this can be found in (McFarland 2001).

54　See (Webster 2009, pp. 42–79).

As I have indicated in a previous work, this form of generic universalism[55] is one that seeks to mask the presumption that the default positionality in how we understand humanity is predicated on Whiteness.[56] My critique is of Whiteness as macro epistemological framework for privilege and superiority, sometimes even triumphant supremacy. This first conversation is principally about the development of ideas and discursive practices as opposed to the subjectivities and positionalities of White people per se. I am not arguing that every White person is imbued with either power or privilege, particularly of the economic kind in terms of the latter. However, what I am interested in and trying to assert in this exploratory paper are the ways in which Whiteness operates as a tacit, concealed form of normative framing, what we have come to know as truth, especially that which contains a universal posture to its ideological claims for itself.

In more recent times, a branch of scholarship entitled "Critical White Studies"[57] has begun the task of naming and unmasking the privileged construct that is "Whiteness". The power of this process, if undertaken with due consciousness to exposing the truth, is one that will be an unflinching and unsparing critique of the hypocrisy of White Christianity, particularly that which has emerged from Euro-American evangelicalism.

The notion of the church as a body that is united under the Lordship of Jesus Christ is one of the enduring truths of the Christian faith. This sense of unity that is so boldly proclaimed as central to the self-understanding of the church itself has often proved more illusionary than real. While the Body of Christ has been fractured by arguments over doctrine, denomination, and issues of class, gender, and sexuality, perhaps the most ongoing challenge and indeed the most persistent scourge has been that of racism.

Our present epoch has witnessed the continued idolatrous nature of aspects of White American evangelicalism, for example, that it has always preferred worshipping White supremacy than the Jesus who tells us to love our neighbours, irrespective of ethnicity or culture or "race", including those who are Black. The recent upsurge in White nationalism in the US in the wake of the Presidency of Donald J Trump and his excoriating rhetoric aimed at those who are deemed the Other is a sad reminder of the singular importance of this paper.

The continued growth of White nationalism across the world is a reminder of the ongoing challenge of resisting White supremacy. In Britain, we have had to deal with our own experience of White supremacy. The Brexit vote clearly demonstrated the barely concealed exceptionalism and sense of entitlement of predominantly White English people.[58] The clear xenophobia underpinning the Leave campaign reminded many of us that "True Britishness" equals Whiteness and that those who are deemed the "Other", be it "migrants" living in the UK or "foreigners" from Europe, are distinctly less deserving in the eyes of many White British people. It can be argued that the romantic push for the nostalgia of the past (when Britain had the biggest empire the world has ever seen) is predicated on the intrinsic value of Britain being superior to others, often seen in terms of groups such as "Britain First" or other groups on the political right who want to "make Britain great again". To quote the Black British social commentator Gary Younge, "not everyone, or even most of the people who voted Leave were driven by racism. But the Leave campaign imbued racists with a confidence they have not enjoyed for many decades and poured arsenic into the water supply of our national conversation."[59]

It is important to note the contested nature of the aforementioned discourse, often focusing on the divergent positionality of Black and Postcolonial scholars when juxtaposed with more centrist

---

[55]　By this I mean how White authors write and speak in an alleged universal language and whose work then has universal applicability. See (Reddie 2006a, pp. 46–51).

[56]　See (Reddie 2003, pp. 68–70, 142–46).

[57]　Some of the key texts in this emerging discourse: See (Perkinson 2004). See also (Perkinson 2005; Cassidy and Mikulich 2007; Harvey 2007). See (Harvey 2014).

[58]　See (Reddie 2019, pp. 1–37, 89–109).

[59]　For an insightful left-wing critique of Brexit that challenges class based notions of privilege and explores notions of White entitlement and racism, see (Younge 2016).

ones seeking to offer a more emollient reading of our post-Brexit milieu. The latter, often writing from appreciative perspective of seeking to offer ameliorative hermeneutics for the rise in White British nationalism, will argue that the rise in intolerance to the conspicuous socio-cultural difference of multiculturalism and immigration can be understood more in terms of fear and alienation than in the frameworks afforded by race analysis as adopted by scholars such as myself.[60]

Clearly, the rise of predominantly White British nationalism has many facets, and I am in no doubt that fear, alienation, and dissatisfaction with the current socio-political milieu that is modern Britain had an important role to play in why so many poorer White people in the former industrial heartlands of the traditional Labour Party voted for Brexit. However, to assert that the resurgence of nationalism has nothing to do with notions of race or White exceptionalism stretches all the normal boundaries of credulity. Let me end this section with a recent conversation I had with my 87-year-old father who lived in Britain from October 1959 until August 1991[61]. He came as a part of the Windrush Generation. He worked in factories in his entire time in the UK. He was an ardent trade unionist, a member of the General and Municipal Workers Union or the GMWU. He was a shop steward for several years and a works convener. When I informed him of the prominent White Anglicans "speaking up for ordinary working-class people" who voted for Brexit,[62] my father retorted, "Why is that when people talk about the working class in Britain, they only mean White people? I worked for over 30 years in Britain in a factory and was a member of a trade union. I was working class. But I bet these people are not standing up for me?"

## 6. An Anti-Racism Ethic in Practice

The existence of racism in Britain today, as we speak, is testament to the continued legacy of Mission Christianity that has always caricatured Africans as "less than" and "the Other", i.e., not one of "us" in terms of the construct of Whiteness. Slavery is long gone, but anti-Black racism has long outlived the institution that helped to breathe it into life. In our contemporary era, the underlying framework of Blackness, which is still symbolically seen as representing the problematic Other, now finds expression in a White police officer placing his knee on the neck of a Black man; despite the plaintive pleas of "I can't breathe", the police officer remains unmoved and maintains his violent posture until this Black man dies. One cannot understand the futility of this death unless you understand that this is no new phenomenon. White power has viewed Black flesh as disposable for the past 500 years.

For those who want to believe that such events as the death of George Floyd could never happen in the UK, let me recall the death of Clinton McCurbin, an African Caribbean man who died of asphyxia at the hands of the police in Wolverhampton on the 20th February 1987, having been arrested for using a stolen credit card.[63] Eye witness accounts spoke of seeing McCurbin gasping for breath as White officers pinned him to the floor and crushed the air out of his body. Later that year, despite the cautionary words from my very law-abiding, hyper-religious, and respectable parents to focus on my studies, I nevertheless travelled to Wolverhampton along with thousands of others to protest the death of Clinton McCurbin. That was my very first march. No officers were ever charged with his death. Life in my local church continued without any recourse to the death of a Black man of dubious

---

[60]　For an excellent example of this, see (Chaplin and Bradstock 2020), where a number of the contributors offer supportive accounts of the rise of White English nationalism (skirting over the rise in racist attacks and xenophobia) in terms of the sense of displacement, fear, and sense of being "left behind" felt by many White working class communities in the Britain. See also (Girma 2018, pp. 117–33; Nixon 2020).

[61]　In 1991, he retired early on health grounds, and he and Mother returned to Jamaica to live in retirement. My mother died in February 2014.

[62]　Of particular note are the chapters by Philip North "Brexit: competing visions of nation", pp. 9–18, and Sam Norton, "Patriotism and theology will have to come together again: Royal Consciousness and the Church of England", whose work argues in defence of White working-class people. At no point do either of them identify Black and Asian migrants as also belonging to the working class and having also suffered from economic deprivation. Both essays are to be found in (Chaplin and Bradstock 2020).

[63]　For further details on the death Clinton McCurbin see (Flash and Hyatt 2020).

character. I cannot recall many occasions in which racism within the church and the wider society was ever addressed in the Methodist church I attended at the time.

It is my contention that succeeding generations of Ordained Methodist ministers and "local preachers"[64] have been trained to ignore the social reality of racism. Whilst previous iterations of the training process for preachers have stressed the importance of social justice as a key element in the Methodist kerygmatic tradition, I am convinced that little exists in the way of an explicit anti-racist ethic that asserts the necessity of deconstructing the toxicity of White normality, entitlement, and privilege.

This brings me to the curious case of the toppling of the statue of Edward Colston in Bristol as part of a Black Lives Matter protest on the 8th June this year.[65] It can be argued that the pulling down or removal of statues has become a distraction against the wider issues of systemic racism that need to be addressed more than the removal of historic artefacts often ignored by most people in their daily activities. That is correct if the focus is solely on statues in and of themselves. However, let us consider the point of the Black Lives Matter Movement in first place.

The Black Lives Matter movement emerged in order to counter the patently obvious fact that Black lives do not matter.[66] This is not just a question of economics or materiality; it is also about seemingly "ephemeral matters" like the impact on our psyche and associated questions of representation and spirituality. It has been interesting observing the concern of many White Christians for the ethical matters of law and order, governance, and property when it comes to the tearing down of the Colston stature in Bristol. Black people, many of whom are the descendants of enslaved peoples, have lived in that city with the sight of a statue built in honour of a slave trader. Polite petitions to move these and other statues were ignored. Long before a so-called mob tore this one down, activists asked for it to be moved to a museum where those who deliberately wanted to see it could while saving those of us who did not the ignominy of having the lives of our oppressed ancestors constantly insulted. White authority ignored our claims, because Black lives and our resultant feelings do not matter. Black lives do not matter in the face of White complacency and disregard. Just as our pleas for justice for Clinton McCurbin went unheeded, because our feelings did not matter either.

Therefore, I find it interesting that following the pulling down of a statue, we had the usual furrowed brow of some White Christians sharing their ethical concern for law and order and the dangers of mob rule.[67] One wonders how many of these complainants were supportive of BLM prior to its sudden resurgence since the death of George Floyd? For some respectable White Christians, their ethical concern is focused on property and not Black lives disfigured by racism. Delroy Wesley Hall speaks of Black people living in Britain struggling with a form of existential crucifixion. We are mired in our continued "Holy Saturday" following our social and collective crucifixion, but with no "Easter Sunday" on the horizon.[68]

Therefore, at this moment in history, I am not going to thank White people for issuing apologies, "taking the knee", writing statements, and going on marches that do not cost them anything when

---

64   Local preachers are non-ordained "lay" people on whom is conferred the authority to preach within the circuit in which they are authorised. For further details on the office of a local preacher, see (The Methodist Church n.d.b.).

65   For details on the pulling down of Edward Colston statue, see (Jannesari 2020).

66   For a helpful distillation of the Black Lives Matter Movement see (Black Lives Matter 2020). See also (Lightsey 2015). Lightsey's work seeks to examine the plural and intersectional nature of the Black Lives Matter Movement, which includes Black LGBTQI+ people, in order to reassert the primacy of all Black bodies mattering and not just respectable, heteronormative, church-going ones.

67   This comment is reflective of the push-back of "some" White Christians on social media responding to the threat to law and order and property. It is important to acknowledge the many Black Christians who have also shared their disquiet at the dangers of mob rule and the desecration of public monuments. I am forced to acknowledge that there are obvious dangers of untrammelled "violent" direct action of this sort. My comments are not an absolute endorsement of this action, but a criticism of the complicity of the authorities in the city to side with the blandishments of White supremacy that is exemplified in the maintenance of statue of Edward Colston in the first place.

68   (Hall 2009).

we are dealing with forms of existential crucifixion that lead to us being more likely to struggle with mental ill health issues such as schizophrenia.[69]

I am not going to "educate" White people on how to deal with their discomfort and emotions when I and countless Black people are afraid to go out of our houses lest we end up as part of the disproportionate numbers who are stopped, detained, and questioned by our supposedly benign police force for violating the changeable rules on social distancing post-lockdown that see White people congregating with impunity.[70]

Thinking back to 1987, when I asked my White Christian colleagues and friends to support me in mounting a campaign to mark the callous killing of Clinton McCurbin, I was met with complete indifference. McCurbin's death did not resonate with them because the death of another anonymous Black man was no big deal. However, every Black person knows that in and of itself, George Floyd's death is not remarkable. Systemic racism did not start with George Floyd's death, nor will it end with White people wringing their hands in liberal guilt, telling us how sorry they are for the racism that blights our lives and not theirs. The bitter truth is that Black lives have not mattered for a very long time, and the Church has long been complicit in this.

I have used the iconic toppling of the Colston statue as a microcosm for the wider Black Lives Matter movement and the indifference of some White Christians to our pleas for justice. The frustration of the protestors that led to the toppling and disposal of the statue reminds me of the very human anger and frustration of Jesus in turning out the money changers in the temple (Matt. 21: 12–17, Mark 11: 15–19, Luke 19: 45–48 and John 2: 13–16). It seems like it is alright for a "White Jesus"[71] as depicted in Western iconography to be angry and destroy property, but not unruly Black people! An anti-racist ethic in Christian ministry is one that most support Black Lives Matter if White Christians are serious about seeking to be in solidarity with Black people as we wrestle with the continued realities of systemic racism.

## 7. Conclusions

This article has sought to outline the development in my own scholarship, ministry, and activism, one that has moved from racism awareness to a theological deconstruction of Whiteness and its relationship to Mission Christianity. This work has also challenged the contemporary predilection for the conceptual framing of unconscious bias and equalities and diversity when juxtaposed with the emphasis on racial justice and anti-racism. Deconstructing Whiteness and its relationship to Mission Christianity represents, I believe, a radical and robust means of developing an anti-racist ethic that can inform and radicalise those training for public, authorised ministry.

I have yet to develop a means of converting this work into a pedagogical framework that can be delivered in a training context for those undertaking ministerial formation in the context of theological education within the "Common Awards"[72] framework. This work is a tentative heuristic that is incomplete. Additional pedagogical and curriculum work is needed for the effective implementation of this proposal for rethinking how churches undertake the task of ordinands for a life of an anti-racist ethic in their public ministry.

Given the neo-colonial construct of patronage on which many churches continue to operate, where epistemological power resides in the hands of White authority figures who are imbued with the referential power of hierarchy, I am not holding my breath that there will be many takers for this

---

69  For an excellent exploration of the ways in Black Christian faith has been an essential means of dealing with the environmental features of systemic racism that leads to disproportionate levels of mental ill health amongst African Caribbean people in Britain, see (Willis 2006).

70  An example of this can be found in (Dearden 2020).

71  For arguably the most comprehensive appraisal for the epistemological frameworks that have given rise to the predominance of a White Jesus, see (Kelley 2002).

72  Common Awards is a Church of England led, but ecumenical, validating mechanism for those for ordained and authored lay ministry in the major Historic churches in England. For more details see (Durham University 2017).

radical theo-ethical approach to anti-racism. Conversely, we are experiencing a seemingly seismic breakthrough in the recognition of systemic racism by hitherto blithely unaware White institutions and their custodians, so we may be witnessing a Kairos moment in the long-awaited breakthrough in repulsing the historic manifestations of racism. As I am not possessed of special powers of prescience, history will be the ultimate judge on whether this is a moment of substantive change or if it simply a momentary blip in the ongoing remorseless march of White supremacy. I will leave the final word to one of the great thinkers on the challenges of converting White people to an ethic of anti-racism, namely, James Baldwin. Baldwin opines: "Not everything that is faced can be changed. But nothing can be changed until it is faced."[73] Are we willing to face the reality of racism?

**Funding:** This research received no external funding.

**Conflicts of Interest:** The author declares no conflict of interest.

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
