# Peer review of "Reassessing the Inculcation of an Anti-Racist Ethic for Christian Ministry: From Racism Awareness to Deconstructing Whiteness"

_religions, doi:10.3390/rel11100497_

Round 1

Reviewer 1 Report

Generally speaking the article is good.  I have a few general recommendations and one specific recommendation.  

In terms of general recommendations, there is a lot going in the paper.  At times, the manuscript reads like a stream of consciousness and feels a bit choppy.  It is not necessary to say everything. Some of the information used to support the author's argument could be removed or moved to footnotes, while the core ideas could be expanded on.  Also, the author jumps in out of personal experience.  This is distracting for the reader, takes away from the authors arguments, and at times sounds a bit like a rant or again a stream of consciousness.  I would carefully select were and how to interject personal experience.  

Specifically, the paragraph beginning with Line 227 - needs clarification.  The phrase  "Christianity is a violent religion" is problematic.  I would concede that expressions, understandings, interpretations etc. of Christianity are violent, but to make this general claim about Christianity is not accurate.  These paragraphs need to ne flushed out a bit more.

Author Response

Thank you very much for your kind review. I have revised the manuscript in red highlights based on your comments. Please see the latest version.

Reviewer 2 Report

The author's excellent work is to be highly commended. They clearly have both a lived and an academic understanding of the issues at hand and are able to blend and move seamlessly between the two in a manner reminiscent of the late, and cited, James Cone.

The author's use of first-person reflection in conjunction with theological research grounds this constructive seeking in its historical moment. While it is possible to view such an article as incomplete, the lack of absolute conclusion should rather be read as intentional: this is a movement toward a new Practical Theology framework for dealing with racism and white supremacy, and, as such, the inconclusive end of the reflection presents a horizon and leasing edge for change.

Author Response

Please see the attached and the new text in red (lines 227 and again at 534) in which I have nuanced and amplified points made by the two referees. As the comments made by the reviewers were relatively brief and not wishing to extend the article unduly, I have tried to provide a concise response. I trust this is satisfactory. 

Reviewer 3 Report

Relevant to the discourse inhabited, this author begins by identifying and contextualizing the social location and starting point for the essay's critical analysis. It also responsibly identifies the methodological approach it seeks, as a dual intersection of practical theology and black theology, and scholarship and ministry. Similarly, by drawing on personal narratives and experiences, it embodies, and thus verifies and validates, its practical theological approach.

Although there is potential to be bogged down early in the definition of terms, on first reading it appeared as if the essay could have set down the parameters and discursive assumptions more clearly at the outset. It states that its understanding of 'race' (with inverted commas) is meant to suggest--based on established scholarship in the fields of cultural studies, feminist and critical race theories, etc.--that this notion is socially constructed, and, thus, goes hand in glove with racism. However, my initial comment was that it could have been more substantive in its reference to such scholarship, even through footnote 6 (which just says 'a number of scholars' and then lists the CCRJ and two sources), including such influential texts in theology and religion as: Cornel West's Prophecy Deliverance!; J. Kameron Carter's Race: A Theological Account; Willie Jennings' The Christian Imagination; Kelly Brown Douglas' Stand Your Ground, etc. In other words, it seemed as if the essay sought to move beyond this point quickly, by building on it as a jumping off point, and thus bolstering the reference to relevant scholarship would have helped anchor it more steadily. Doing so would also make footnote 10 redundant.

However, after finishing the entire article, the question and initial comment shifted, because the essay does return to what (at first glance) appeared to be its initial 'premise' concerning 'race,' and in so doing weaves in more substantive engagement with scholarship. For instance, on page 7 (beginning on line 205), the essay explicitly mentions Jennings' work on 'race' and how the author has been influenced by this work. Similarly, on page 8 (in footnotes 28 and 30), the author draws on Carter and Douglas, respectively. The question then became for me as a reader, did the essay intend to develop its understanding of 'race' as part and parcel of its argument, as opposed to building on what it seems to present as an initial premise (which was my first thought)? If the latter, which appears more convincing to me, then the quickest fix (in this reviewer's opinion) would be to state that more clearly at the outset. In other words, to avoid the confusion that this reader fell into, clarify that part of the argumentative thrust of the essay will be to demonstrate how and why a certain understanding of 'race' functions in the way it does in practical ministry training.

Overall, however, the essay's main point--that previous ministerial training on racism is flawed because of its 'failure to name the White, privileged, androcentric inherited norms that have underpinned Christianity since the epoch of colonialism' (p.11)--offers not only a sound and sober warning, but one that is significant and relevant for the contemporary moment. As is implied in the article, since the killing of George Floyd and ensuing protests globally, more corporations and institutions--secular and religious--have begun initiating and/or scrambling to resuscitate diversity and inclusion practices and policies. Thus the essay's challenge to previous attempts functions as a sober warning to contemporary practices. In so doing, it offers fresh insight and analyses that break new ground and build upon current scholarship.

Author Response

I have responded to the third/latest peer reviewer by adding an additional paragraph to better explain the premise and the parameters of the work. I have also augmented an early footnote to match this change and then removed a later one (footnotes 6 and 10 respectively) as suggested by the reviewer. I am indebted to them for their kind insights re: the earlier section of the work. 

All the changes pertaining to the three reviewers have been highlighted in red. 

Round 2

Reviewer 3 Report

Excellently revised - happy for publication with minor editing for spell check.